# Simulation and Experimental Study on Sealing Characteristics of Hydro-Pneumatic Spring GS Seal Rings

Shuai Wang [1,2], Pengyuan Liu [1], Donglin Li [1,2], Zhenle Dong [1,2] and Geqiang Li [1,2,*]

1    School of Mechatronics Engineering, Henan University of Science and Technology, Luoyang 471003, China; zldong@haust.edu.cn (Z.D.)
2    Henan Collaborative Innovation Center for Advanced Manufacturing of Mechanical Equipment, Luoyang 471003, China
*    Correspondence: hitligeqiang@163.com; Tel.: +86-151-3873-9516

**Abstract:** Hydro-pneumatic springs often work under high pressure and alternating load conditions, which can easily cause seal damage, leakage, and serious failure. Relevant studies have shown that the dynamic seal failure of hydro-pneumatic springs is the main cause of hydro-pneumatic spring failure, and the sealing performance of GS sealing rings directly determines the service life of hydro-pneumatic springs. The influence of different sealing structure parameters on the sealing performance of GS sealing of hydro-pneumatic springs has been studied, which can provide effective theoretical guidance for the design of sealing structures related to hydro-pneumatic springs. However, the existing research on hydro-pneumatic springs mainly focuses on the characteristics of damping and stiffness; there is a relative lack of research on sealing performance and sealing structure, and the performance change law of GS seals under different working conditions is unclear. In this paper, a GS seal ring is selected as the main seal of the hydro-pneumatic spring, the material parameters of the GS seal ring are obtained via the single-axis compression test, and the finite element simulation method is used to establish the sealing model under different compression ratios of 10–30% and different pressure impacts of 5–30 MPa. By doing so, the stress nephogram, sealing ring shape, and sealing contact pressure of the GS sealing ring under different simulation parameters are obtained. From the test results, the decrease in compression ratio after the wear of the sealing ring is the main reason for the seal leakage of the hydro-pneumatic spring. The maximum contact stress of the sealing ring occurs at the lip of the step ring, and the maximum sealing pressure of the sealing ring is determined by the contact pressure of sealing surfaces II and III. The sealing performance of the GS-type combination sealing ring is affected by the compression ratio of the sealing ring and the impact pressure; when the compression ratio of the sealing ring is 15%, the sealing ring can meet the sealing work needs below 25 MPa. The research results provide a theoretical basis for the design of GS sealing of hydro-pneumatic springs and the effective improvement of the life and reliability of related equipment.

**Keywords:** hydro-pneumatic spring; seal; pressure impact

## 1. Introduction

Hydro-pneumatic suspension is widely used in military vehicles, mining trucks, and construction machinery vehicles due to its nonlinear variable stiffness characteristics and nonlinear damping characteristics [1,2]. The hydro-pneumatic spring is the core component of hydro-pneumatic suspension; it uses inert gas as the elastic medium, with oil to transmit pressure, and through the compressibility of elastic gas and the damping characteristics of hydraulic oil, adapting to the various use conditions of the vehicle [3–6], the reciprocating linear movement of hydro-pneumatic springs and high-pressure impact imposes higher requirements on the sealing structure. The sealing form of a hydro-pneumatic spring is mainly divided into static and dynamic seals: the static seal is composed of an O-ring and

retaining ring, located between the end cover and the cylinder barrel; the dynamic seal consists of a GS sealing ring, which is located between the end cap and the piston rod. The structure of the GS sealing ring is similar to the structure and principle of the Steseal seal and has the advantages of low friction, no crawling, a small starting force, high-pressure resistance, and a simple groove structure [7].

Hydro-pneumatic springs often work under high pressure and alternating load conditions, which easily cause seal damage leakage and lead to serious failure, seriously affecting vehicle performance and even causing safety accidents. Relevant studies have shown that the dynamic seal failure of hydro-pneumatic springs is the main cause of hydro-pneumatic spring failure, and the sealing performance of the GS sealing ring directly determines the service life of a hydro-pneumatic spring [8]. Therefore, studying the sealing performance change law of GS sealing rings under different working conditions can provide an effective theoretical basis for the design of the sealing structure of hydro-pneumatic springs. At present, many scholars have conducted in-depth research on hydro-pneumatic springs, but most of these studies have focused on the damping and stiffness characteristics of hydro-pneumatic springs [9]; there is a relative lack of research on the sealing performance and sealing structure of hydro-pneumatic springs, and the design of related sealing structures is mainly based on experience and lacks theoretical guidance. Gui, P. [10,11] established the mathematical model of hydro-pneumatic spring fluid, derived the leakage flow formula of Steseal, and, combined with the finite element simulation method, analyzed the factors affecting the leakage amount of the Steseal seal and the main reasons for the seal failure of the hydro-pneumatic spring from the aspects of seal lubricity, temperature range adaptability, and oil–fluid compatibility based on engineering practice experience. Yang, C. [12,13] analyzed the main failure modes and mechanisms of hydro-pneumatic springs, proposed a fault identification and prediction framework for the faults of air leakage, and verified the feasibility of the method with experiments. Some scholars have chosen a hydraulic actuator, which is similar to the hydro-pneumatic spring reciprocating seal structure, as the object of their research. Ganlin, C. [14] found through a comparative analysis that the main reason for the actuator seal failure is the wear of the sealing ring; the wear of the sealing ring leads to a rapid decrease in contact pressure and sealing performance. Ran, H. [15] studied the influence of actuator piston rod texture and sealing pressure on the wear of the sealing ring via numerical simulation and found that the texture of the piston rod surface had a significant effect on fluid pressure, film thickness, and contact pressure. Through the simulation and analysis of the VL sealing structure of an aviation actuator, Ouyang, X.P. [16] obtained the contact pressure distribution of the sealing ring under different fluid pressures and revealed the changing relationship between the sealing structure and the contact pressure distribution during the process of fluid pressure establishment. From the above, there are few studies on hydro-pneumatic spring seals, and the performance change law of GS seals under different working conditions is unclear. Although there is some research on the reciprocating sealing technology of actuators, the impact load sustained by hydro-pneumatic springs during operation is lon, and the vibration frequency is fast. There is a significant difference in the reciprocating sealing technology used by the two, and the relevant research conclusions are difficult to directly apply to the sealing design of hydro-pneumatic springs. Therefore, it is necessary to study the influence of different sealing structure parameters in the sealing performance of hydro-pneumatic spring GS, to provide a reasonable theoretical basis for the design and selection of related structures.

Given the above problems, this paper takes YQHF-CC180/150*200B (Shandong Wantong Hydraulic Co., Ltd., Rizhao, China) hydro-pneumatic spring products as the research object and analyzes the contact pressure and stress–strain properties of the sealing surface of the GS15006 sealing ring with a compression ratio of 10–30% under the impact pressure of 5–30 MPa based on finite element simulation. A hydro-pneumatic spring test bench was built, and GS sealing rings with different compression ratios were selected to test and verify their sealing performance and wear under the corresponding impact pressure, reveal the change law of stress and strain of GS seals under different impact conditions, point out

the stress concentration failure area of GS sealing rings, provide theoretical guidance for the design and improvement of GS sealing structure of the hydro-pneumatic spring, and improve the reliability of related products.

## 2. Materials and Methods

### 2.1. Hydro-Pneumatic Spring Sealing Structure

In practical operation, hydro-pneumatic springs often experience oil leakage and oil seepage at the dynamic sealing area due to the reciprocating motion of the piston rod and the continuous hydraulic pressure impact inside the cylinder. As shown in Figure 1, the YQHF-CC180/150*200B model is a hybrid hydro-pneumatic spring with a cylinder diameter of 180 mm, a rod diameter of 150 mm, and a maximum working stroke of 200 mm. It requires a maximum working pressure of up to 25 MPa, and the working medium used is No. 10 aviation hydraulic oil and nitrogen gas. There are two seals places in the hydro-pneumatic spring: static and dynamic seals. The static seal is located between the end cap and the cylinder; the dynamic seal is a GS15006-type combination sealing ring. This paper studies the sealing performance of the GS15006 sealing ring under different parameters. The GS sealing ring is composed of a step ring and an O-ring. The elastic force of the GS seal is generated by the amount of precompression of the O-ring, which places the step ring close to the sealing surface and plays a key sealing role [17].

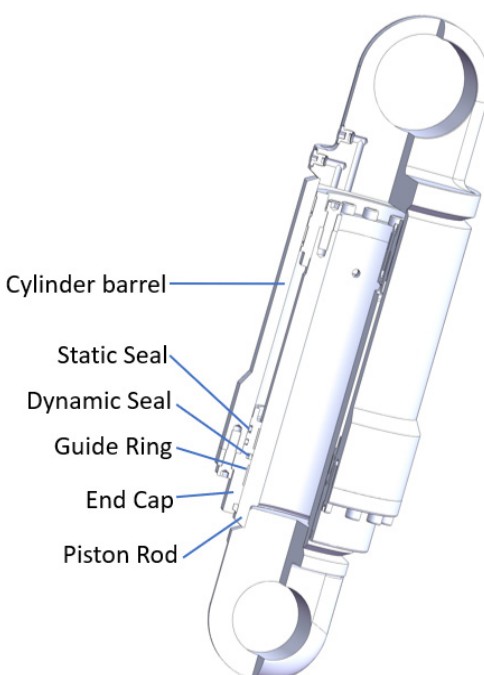

**Figure 1.** Local cross-sectional view of hydro-pneumatic spring.

### 2.2. Sealing Ring Parameters

#### 2.2.1. Geometric Parameters

In Figure 2, a schematic diagram of the installation location of the GS seal ring is presented. The cross-sectional geometric dimensions of the step seal ring are illustrated in Figure 3a, while the cross-sectional geometric dimensions of the O-ring are shown in Figure 3b.

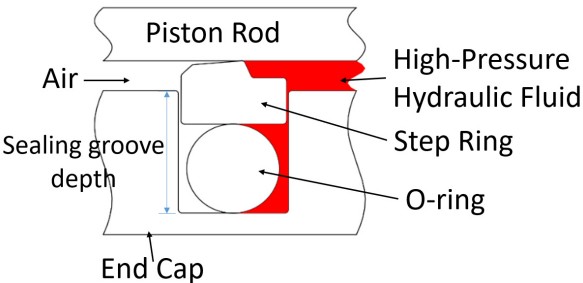

**Figure 2.** GS seal ring.

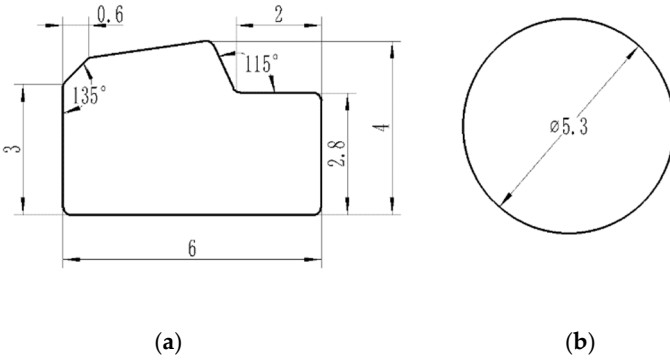

(**a**)                              (**b**)

**Figure 3.** Geometric dimensions of GS seal ring cross-section (unit: mm): (**a**) step seal ring, (**b**) O-ring.

### 2.2.2. Determination of Material Parameters

The GS seal ring consists of a step seal ring and an O-ring. The step seal ring is made of copper-filled polytetrafluoroethylene (PTFE), and the O-ring is made of modified nitrile rubber (NBR). According to the requirements of GB/T 1041-1992 [18] 'Test Method for Plastic Compression Properties', cylindrical specimens of copper-filled PTFE with a diameter of 12 mm and a length of 30 mm were prepared, totaling 5 specimens, and compressed at a speed of 1 mm/min. Following the requirements of GB/T 7757-2009 [19] 'Determination of Compression Stress–strain Properties of Sulfur-Cured Rubber or Thermoplastic Rubber', cylindrical specimens of nitrile rubber rod with a diameter of 29 mm and a length of 12 mm were prepared, totaling 3 specimens, and compressed at a speed of 10 mm/min. The test materials were supplied by the seal ring manufacturer. Figure 4a shows the PTFE specimen, and Figure 4b shows the NBR specimen.

The specimens were subjected to uniaxial compression tests using the WAW-600D (SUNPOC$^{\text{TM}}$, Shenzhen, China) servo-hydraulic universal testing machine, as shown in Figure 5. The stress–strain curves of the specimens were obtained. By analyzing the stress–strain curves of five PTFE specimens, the average elastic modulus of PTFE was calculated to be 292 MPa, with a Poisson's ratio of 0.45. The stress–strain curve test data for NBR were imported into the ABAQUS property module, and a constitutive model for NBR material was fitted using the material evaluation function. In this study, the Mooney–Rivlin hyperelastic model was chosen, with a strain energy density function as follows:

$$W = C_{10}(I_1 - 3) + C_{01}(I_2 - 3) + \frac{1}{2}D(I_3 - 1)^2 \tag{1}$$

In the equation, $W$ represents the strain energy density function, $I_1$ and $I_2$ are the stress tensor invariants, and $I_3$ and $D$ reflect the material's deformation behavior. $C_{10}$, $C_{01}$, and $D$ are material constants, which were calculated based on material test data imported into the ABAQUS 2022 software, resulting in $C_{10}$ = 1.183 MPa; $C_{01}$ = 0.479 MPa; $D$ = 0.004.

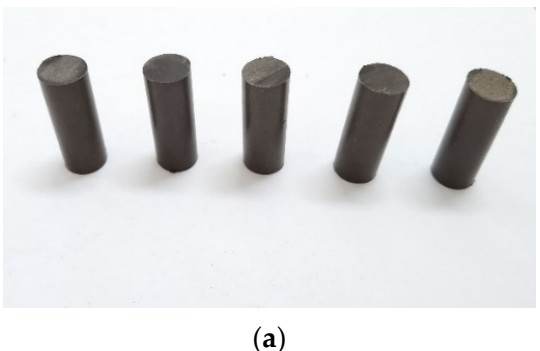

(**a**)

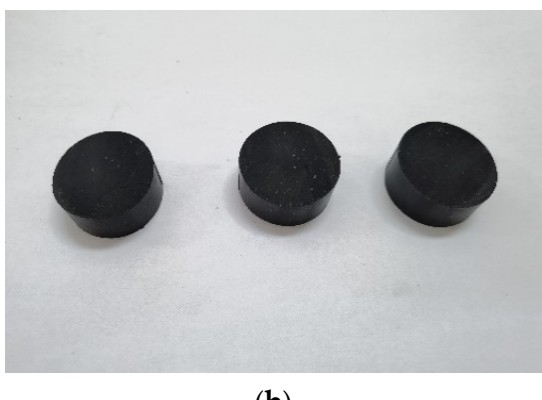

(**b**)

**Figure 4.** Test specimens: (**a**) PTFE specimens, (**b**) nitrile rubber specimens.

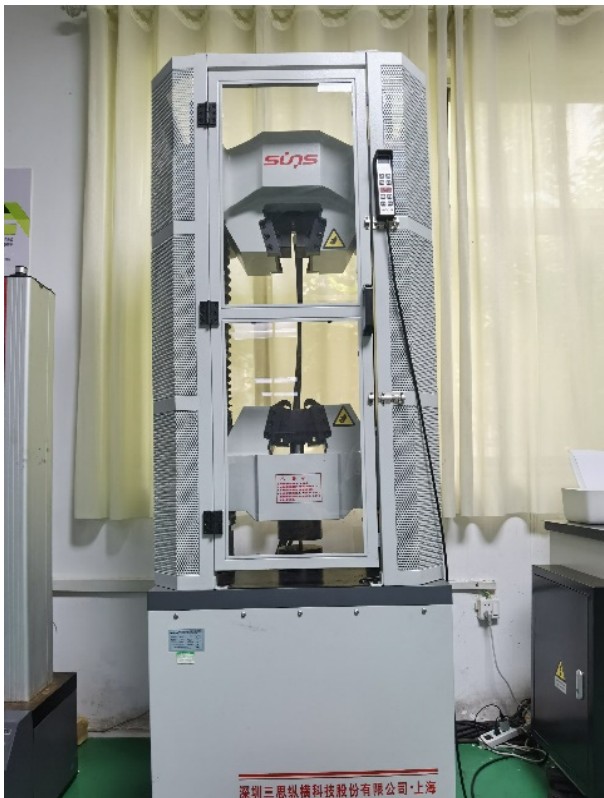

**Figure 5.** WAW-600D electro-hydraulic servo universal tensile testing machine.

*2.3. Finite Element Simulation Model of Hydro-Pneumatic Spring Seal Based on ABAQUS*

NBR is classified as a hyperelastic material, and its mechanical behavior under compression deformation exhibits strong nonlinearity. It undergoes complex deformation

after loading, often accompanied by large displacements and strains, which can lead to convergence issues during finite element simulation calculations. Therefore, the following assumptions were made during the finite element simulation analysis:

　　1. Neglecting the influence of temperature on the mechanical properties of rubber material.

　　2. Neglecting the stress relaxation and creep characteristics of rubber material.

　　ABAQUS software is proficient in solving nonlinear problems, and its excellent solver performance, efficient simulation of complex contacts, and capability to automatically fit rubber test data and optimize constitutive models are advantageous for addressing various highly complex nonlinear problems.

### 2.3.1. Model Establishment

　　A 2D axisymmetric model for the step seal ring, O-ring, hydro-pneumatic spring piston rod, and end cap was established based on the cross-sectional geometric dimensions of the GS seal ring, as shown in Figure 3a,b. Considering that the elastic modulus of the cylinder barrel and piston rod is significantly greater than that of PTFE and NBR, the cylinder barrel and piston rod were set as rigid bodies, and material properties were defined and assigned to each component before assembly.

### 2.3.2. Meshing

　　The cylinder body and piston rod were treated as analytical rigid bodies and did not require meshing. Meshing was performed only for the step seal ring and O-ring. The step seal ring was meshed using CAX4R elements with reduced integration, totaling 2128 elements. The O-ring was meshed using CAX4RH elements with reduced integration and was segmented to improve convergence during simulation and to define contact surfaces. The meshing of the O-ring consisted of a total of 2580 elements, as shown in Figure 6.

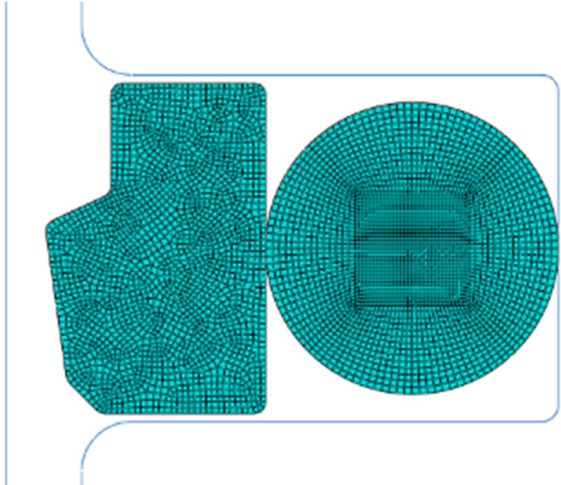

**Figure 6.** The meshing of the seal rings.

### 2.3.3. Boundary Condition Settings

　　There were four contact interfaces in total. Sealing surface I: step seal ring with the hydro-pneumatic spring piston rod. Sealing surface II: step seal ring with the O-ring. Sealing surface III: O-ring and the groove on the hydro-pneumatic spring end cap seal. Contact surface IV: step seal ring with the groove on the hydro-pneumatic spring end cap seal. Among these, Sealing surfaces I to III represent pressure penetration surfaces, as shown in Figure 7.

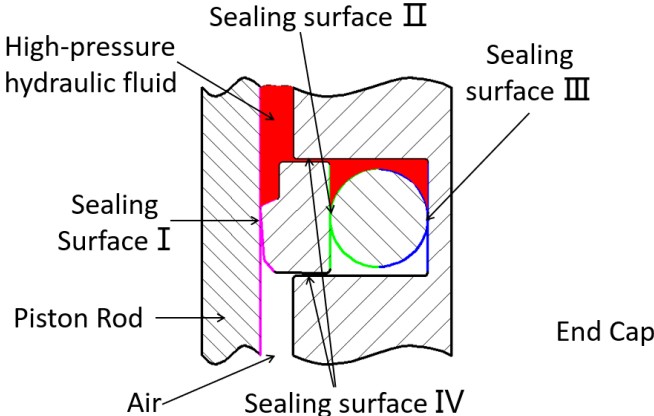

**Figure 7.** Pressure penetration schematic.

2.3.4. Solver Settings

The analysis was divided into three steps:

① Solving for the contact between the seal rings;

During assembly, there is an overlap between the O-ring and the end cap groove, as well as between the step seal ring and the O-ring. The software automatically finds the contact surfaces using the interference adjustment command in the interaction module of ABAQUS software.

② Fixing the end cap and compressing the seal rings with the piston rod;

The compression ratios of the entire seal ring are chosen as 10%, 15%, 20%, 25%, and 30%. The installation gap between the piston rod and the end cap is set to 0.2 mm. The calculation formula of the compressibility *W* is as follows:

$$W = (d_0 - h)/d_0 * 100\% \tag{2}$$

where $d_0$ is the initial radial length of the seal before compression, mm; $h$ is the radial length of the compressed rubber parts, mm. The compression ratio of the seal ring is controlled by changing the depth of the seal groove arranged on the end cap. The displacement of the piston rod and the depth of the seal groove for different compression ratios are shown in Table 1.

**Table 1.** Piston rod displacement and seal groove depth for different compression ratios.

| Solution | ① | ② | ③ | ④ | ⑤ |
|---|---|---|---|---|---|
| Compression ratio | 10% | 15% | 20% | 25% | 30% |
| Displacement (mm) | 1.63 | 2.095 | 2.56 | 3.025 | 3.49 |
| Sealing groove depth (mm) | 8.17 | 7.705 | 7.24 | 6.775 | 6.31 |

③ The cyclic pressure shock of high-pressure hydraulic oil on the seal ring.

The pressure infiltration module (penetration pressure) in ABAQUS software can replace complex fluid–structure interaction models to study the fluid loading conditions between two surfaces under the impact of high-pressure liquid. During the reciprocating sealing process, the addition of high-pressure hydraulic oil on one side of the seal ring exerts a force on the seal ring, ultimately transforming into contact pressure at the sealing lip. In finite element analysis, there are generally two methods for applying fluid pressure. The specified boundary method is to pre-set the fluid pressure on the sealing surface before applying the pressure load. However, this method does not automatically adjust the pressure loading surface with the deformation of the seal ring, which can lead to errors in the analysis. The pressure penetration method: this method involves setting up contact pairs for fluid penetration, specifying the starting point for loading, and defining the pressure values to apply. It allows the nodes where fluid pressure is applied to adjust

automatically with the deformation of the contacting bodies, providing a more accurate simulation of the seal ring's deformation process [20].

To realistically simulate the seal ring's response to pressure shock caused by high-pressure hydraulic oil in the working conditions of the hydro-pneumatic spring, a simulated fluid pressure with a frequency of 2 Hz was applied to the seal ring. This analysis aimed to assess the sealing performance between the seal ring and the end cap and piston rod interface under cyclic pressure shock conditions. As shown in Figure 8, the pressure shock cycle included pressure values ranging from 5 to 30 MPa with a 5 MPa increment; the peak value of pressure value is "P".

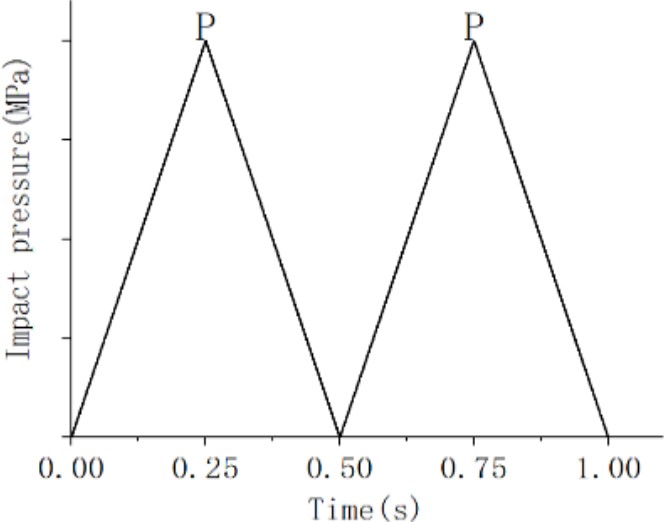

**Figure 8.** Pressure impulse cycle.

### 2.4. Experimental Setup and Testing

The hydro-pneumatic spring test platform, as shown in Figure 9, mainly consists of three parts: the test platform body, the hydraulic pump station, and the control cabinet. To fit the actual use, the hydro-pneumatic spring is connected and installed on the test bench with a pin shaft. The control cabinet is programmed to control the hydraulic system, to realize the reciprocating movement of the hydro-pneumatic spring driven by the hydraulic cylinder on the test bench. The test platform can detect the sealing condition of the hydro-pneumatic spring's sealing ring at different compression rates and assess the wear of the sealing ring after testing.

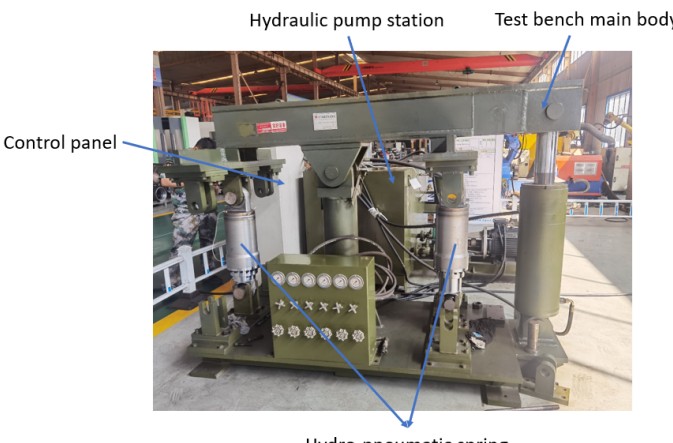

**Figure 9.** Hydro-pneumatic spring test bench.

In practical usage, the hydro-pneumatic spring can be divided into two operating conditions: (1) When the vehicle is stationary, the high-pressure hydraulic fluid inside the hydro-pneumatic spring does not subject the sealing ring to high-pressure impacts. In this case, the sealing ring is subjected to static pressure loading. (2) When the vehicle is in motion, especially on uneven road surfaces, the hydro-pneumatic spring undergoes reciprocating motion, and the sealing ring starts to experience pressure impacts. To address these two scenarios, and based on the conclusions obtained from simulations, the following experimental studies were conducted.

The experimental verification was divided into two parts. The first part verified the sealing conditions of the sealing ring when the compression rates were approximately 10% and 15% under static loading conditions of the hydro-pneumatic spring. The second part verified the sealing conditions of the sealing ring when the compression rates were approximately 10% and 15% under dynamic loading conditions of the hydro-pneumatic spring. These experiments were aimed at validating the sealing performance of the sealing ring at different compression rates, as determined in the simulation. It is important to note that due to machining errors during the actual manufacturing of the sealing grooves, there may be slight discrepancies in the actual compression rate of the sealing ring.

### 2.4.1. Static Loading Test

The components of the hydro-pneumatic spring were cleaned and assembled, a quantity of No. 10 aviation hydraulic oil was injected into the hydro-pneumatic spring, and then the hydro-pneumatic spring was filled with nitrogen at a pressure of $7 \pm 0.2$ MPa. We fixed the hydro-pneumatic spring to the hydro-pneumatic spring test bench and let it sit for an hour. The hydro-pneumatic spring test bench did not need to carry out loading movement; this test step was only used to simulate the sealing performance of the hydro-pneumatic spring seal ring under static compressions of 10% and 15%.

### 2.4.2. Dynamic Loading Test

Using the two hydro-pneumatic springs from the static loading test, the hydraulic system of the hydro-pneumatic spring test bench was set with loading action instructions. The test was divided into five groups, with each group consisting of 1000, 2000, 3000, 4000, or 5000 reciprocating loading cycles. After each group of loading tests was completed, the GS sealing ring was removed and inspected for wear, and measurements were taken. The loading action consisted of two steps:

(1) The action frequency was 1 Hz with an amplitude of 100 mm, completing 50 reciprocating loading actions. This step was intended to ensure adequate lubrication of the sealing ring.

(2) The action frequency was 2 Hz and the amplitude was set to the maximum working stroke of the hydro-pneumatic spring, which is 200 mm. According to the grouping mentioned above, reciprocating tests ranging from 1000 to 5000 cycles were conducted. After each group of tests was completed, the condition of the sealing ring was assessed.

## 3. Results

### 3.1. Analysis of the Sealing Ring Compression Process

By analyzing the deformation and stress of the seal ring under different compression rates, the stress concentration position of the seal ring during operation could be obtained, and the influence of the compression rate on the shape of the seal ring could be judged. After completing the testing of the sealing contact surface in the first step, the second step of the piston rod compression seal ring action was carried out. The stress nephogram of the seal ring corresponding to the completion of the second step of the GS seal ring under different compression rates is shown in Figure 10.

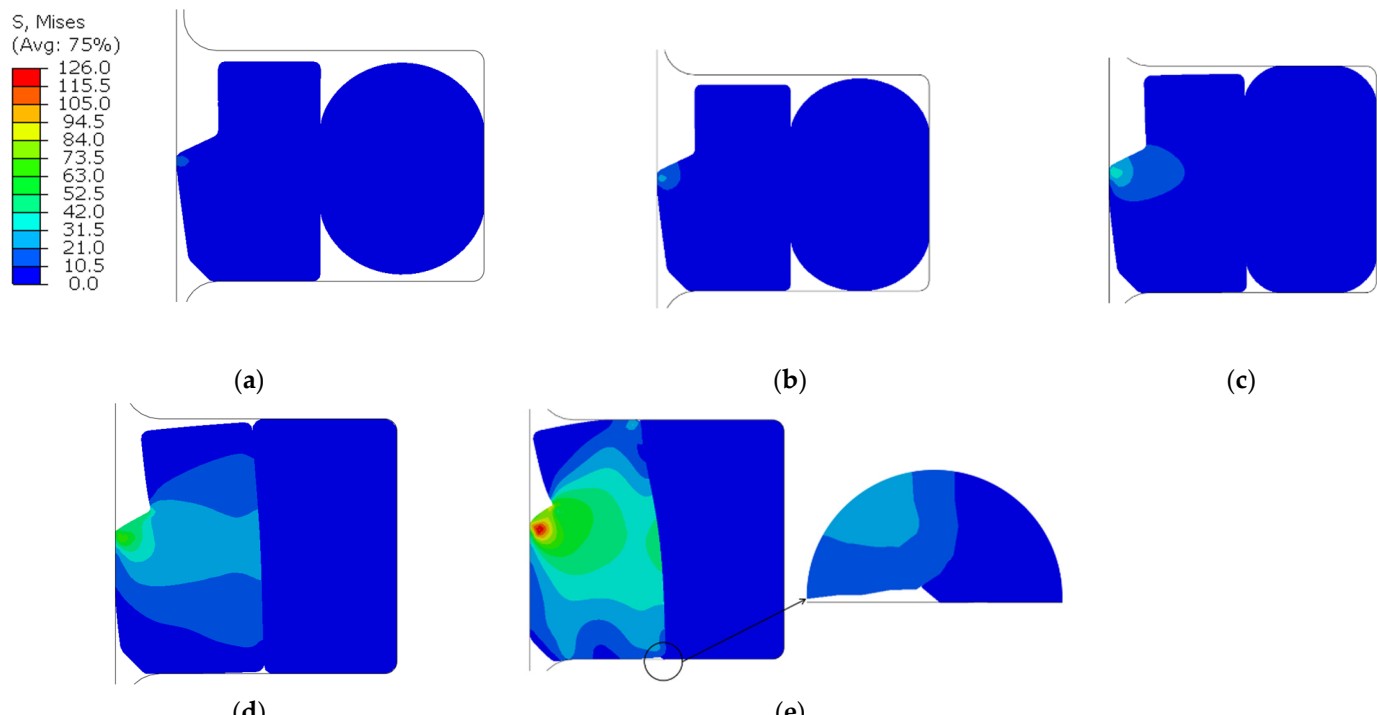

**Figure 10.** Seal stress nephogram at different compression rates (unit: MPa): (**a**) compression ratio of 10%, (**b**) compression ratio of 15%, (**c**) compression ratio of 20%, (**d**) compression ratio of 25%, (**e**) compression ratio of 30%.

According to the stress nephogram in Figure 10, we can observe the following maximum stresses for different compression ratios: compression ratios of 10, 15, 20, 25 and 30% result in maximum stresses of 16, 26, 41, 72, and 126 MPa. From the graph, it is evident that there is a stress concentration at the lip of the sealing ring, which can potentially lead to excessive wear of the lip and result in the loss of sealing performance. At a compression ratio of 30%, the O-ring is completely compressed into the sealing groove and experiences squeezing against the step seal ring, as indicated in Figure 10e. In practical applications, this can lead to shear damage of the O-ring, failing the sealing ring, especially when subjected to shock loads from hydro-pneumatic springs. Excessive stress on the step seal ring can also accelerate its wear and reduce the sealing ring's service life. Therefore, in the subsequent sections, we will not further report the sealing conditions at a compression ratio of 30%.

The path of the contact surface between the step ring and the piston rod was selected, and the contact pressure on the same path under different compression rates was measured. As shown in Figure 11, the contact pressure of sealing surface I under different compression amounts increases with the increase in the compression amount, and the width of the contact surface also increased with the increase in the contact pressure. When comparing the contact pressure of sealing surfaces III and II under different compression amounts, as shown in Table 2, we found that the peak contact pressure of the two sealing surfaces was approximately equal, and the contact width was roughly symmetrical, so the following only analyze the sealing conditions of sealing surfaces I and III.

**Table 2.** Peak contact pressures on different sealing surfaces at various compression ratios (unit: MPa).

| Compression Ratio | 10% | 15% | 20% | 25% | 30% |
|---|---|---|---|---|---|
| Sealing Surface I | 26.96 | 42.93 | 64.28 | 120.78 | 213.47 |
| Sealing Surface II | 3.03 | 4.49 | 6.88 | 14.96 | 34.46 |
| Sealing Surface III | 3.03 | 4.49 | 6.83 | 14.79 | 33.07 |

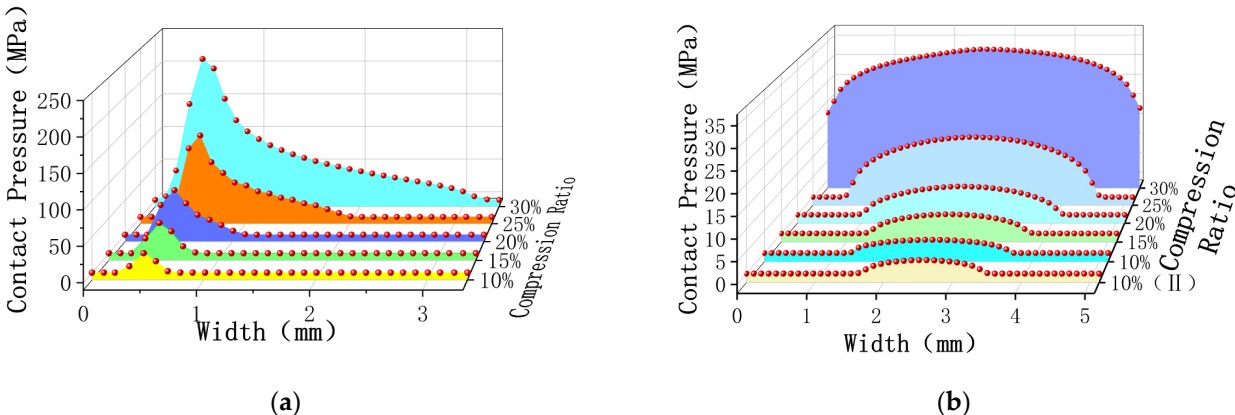

(**a**)          (**b**)

**Figure 11.** Contact pressure on sealing surfaces at different compression ratios: (**a**) contact pressure of the sealing surface I, (**b**) contact pressure of the sealing surface III.

### 3.2. *Analysis of the Pressure Shock Process*

By analyzing the shape, stress, and contact pressure of the sealing ring under the action of impact pressure in the pre-compressed state, we can identify the causes and locations of leaks occurring on the sealing surface. This analysis provides a theoretical basis for the design of sealing structures.

### 3.2.1. Stress Nephogram of Sealing Ring under Different Compression Rates and Different Pressure Impacts

As shown in Figure 12, when comparing the compression of the sealing ring under different compression ratios, we found that the stress of the sealing ring increased with the increase in impact pressure under the same shrinkage ratio; from left to right, the peak impact pressures are 5, 10, 15, 20, 25, and 30 MPa. For example, at a compression ratio of 10%, the maximum stress at the lip of the step seal ring increases from 32 MPa to 72 MPa. At a compression ratio of 15%, the maximum stress at the lip increases from 38 MPa to 80 MPa. At a compression ratio of 20%, the maximum stress at the lip increases from 49 MPa to 88 MPa. At a compression ratio of 25%, the maximum stress at the lip increases from 70 MPa to 109 MPa. It is worth noting that the magnitude of the increase in maximum stress from the 20% to 25% compression ratio is greater than the increase observed at 10% and 15% compression ratios.

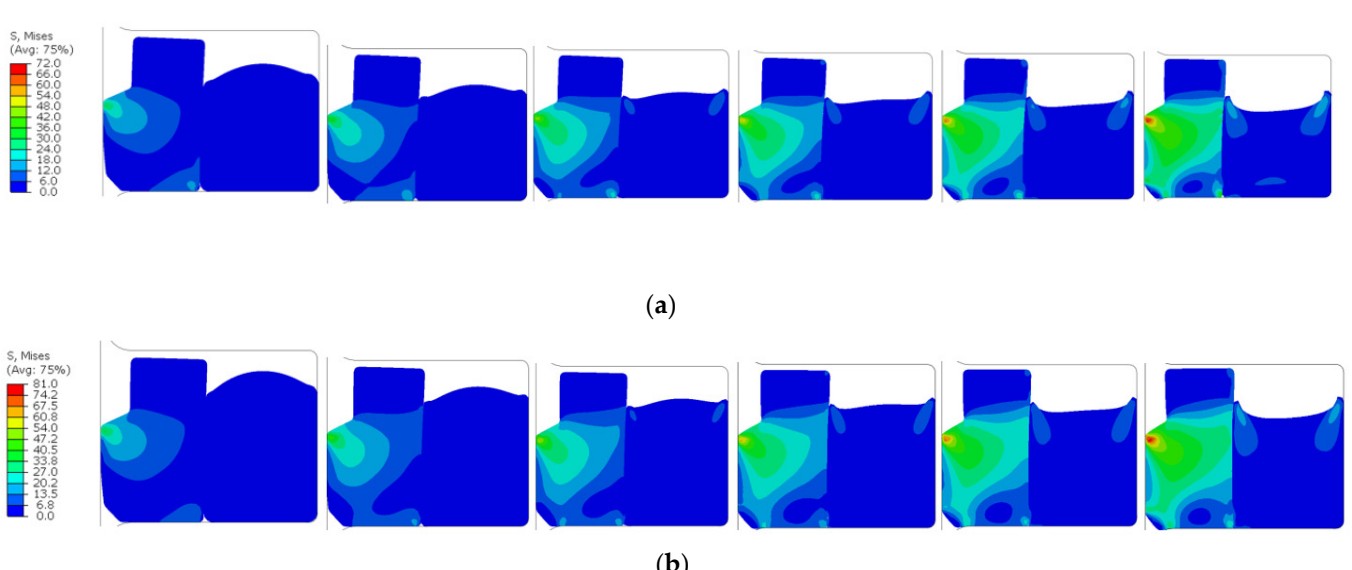

(**a**)

(**b**)

**Figure 12.** *Cont*.

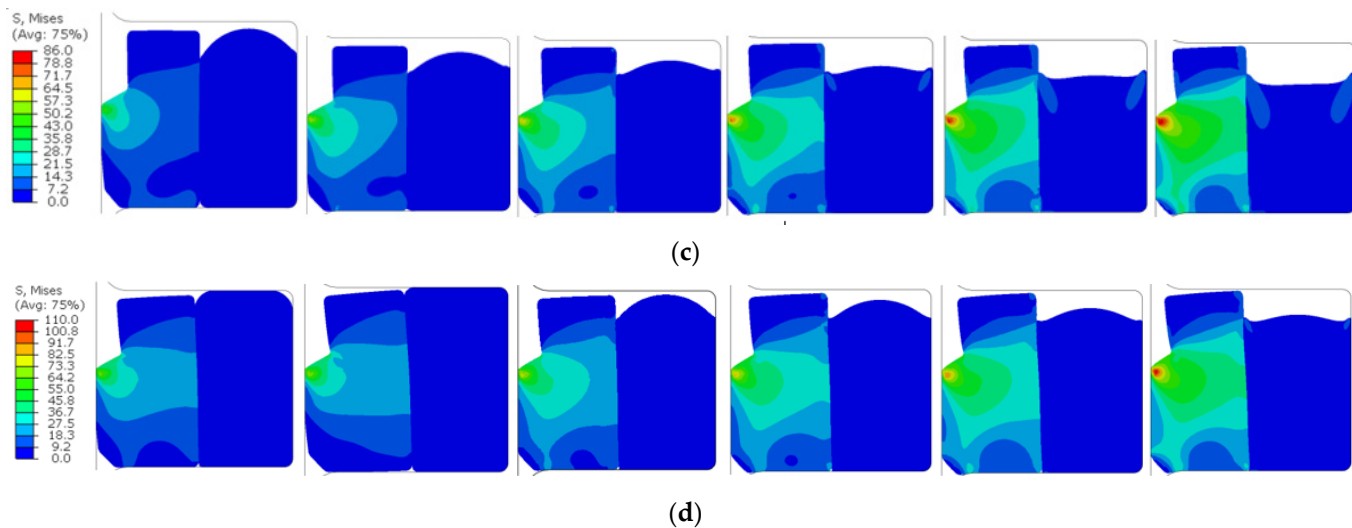

(c)

(d)

**Figure 12.** Stress nephogram of the sealing ring under different compression ratios and pressure impacts: (**a**) compression ratio of 10%, (**b**) compression ratio of 15%, (**c**) compression ratio of 20%, (**d**) compression ratio of 25%.

3.2.2. After the Stress Impact, Stress Nephogram of the Sealing Ring

As shown in Figure 13, the appearances of the seal ring after being impacted by 15 MPa pressure under different compression rates were compared. It was observed that at a compression ratio of 10%, the sealing ring experienced significant displacement between the O-ring and the step seal ring when subjected to pressure impact. Furthermore, it did not return to its pre-impact position after the pressure impact ended. In practical use, significant pressure shocks can cause the sealing ring to shift or even twist, leading to irreversible damage. Therefore, it is important to avoid such occurrences. At a compression ratio of 15%, however, the sealing ring showed a better ability to recover to its initial position after pressure impact, indicating good repeatability and positioning accuracy.

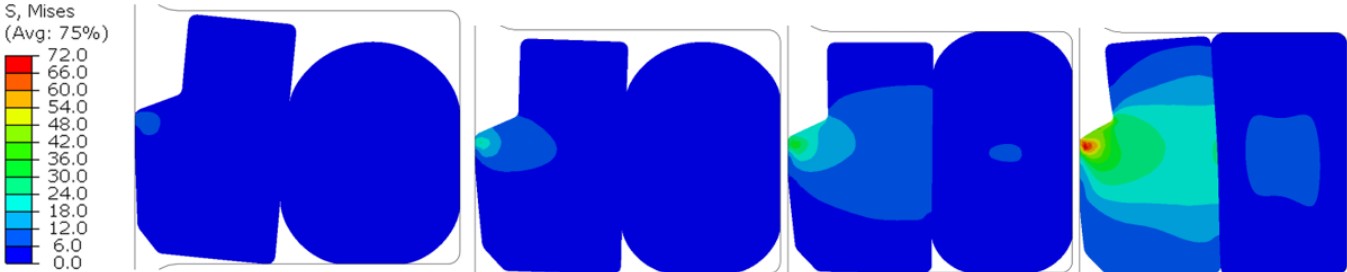

**Figure 13.** Stress nephogram of the sealing ring after pressure impact at compression ratios of 10% to 25%.

*3.3. Contact Pressure on Sealing Surfaces under Impact Pressure*

The necessary condition to ensure the sealing of the sealing ring is that the maximum contact pressure on the sealing contact surface must be greater than or equal to the high hydraulic pressure [21]. By analyzing the contact pressure on the sealing surface of the sealing ring, we could assess whether the sealing performance of the sealing ring meets the requirements. Sections 3.1 and 3.2.2 have already excluded the cases of 30% and 10% compression ratios. Therefore, the following analysis will only focus on the situations where the compression ratios are 15%, 20%, and 25%, concerning the sealing ring's response to pressure impacts.

Simulation data for contact pressures on sealing surfaces under different compression ratios of 15%, 20%, and 25% and varying impact pressures were obtained, as shown

in Figure 14. When comparing the contact pressures on sealing surfaces under different conditions, the following observations can be made: when the sealing surfaces are subjected to pressure impacts, the contact pressure on the sealing surface I increases with the increase in impact pressure, building upon the static compression. Additionally, the contact pressure increases with an increase in compression ratio, and the location of the maximum contact pressure remains approximately the same, occurring at a position roughly 0.5 mm from the edge of the sealing surface. At an impact pressure of 30 MPa, the contact pressure on sealing surface I exhibits a sudden change, and this jump becomes more pronounced with an increase in compression ratio. When the impact pressure is 25 MPa and the compression ratio is 15%, the contact pressure on sealing surface I initially decreases uniformly with the sealing surface. However, when the compression ratio reaches 20% and 25%, the contact pressure on the sealing surface I starts to exhibit a jump, and the jump is more significant at a compression ratio of 25% compared to 20%. When comparing the contact pressure on sealing surface III, it can be observed that, under different compression ratios, the increase in contact pressure with increasing impact pressure is less pronounced compared to sealing surface I. Moreover, the contact pressure on sealing surface III approximates a straight line after the sealing surface is fully established, indicating a relatively uniform distribution without any abrupt changes in pressure.

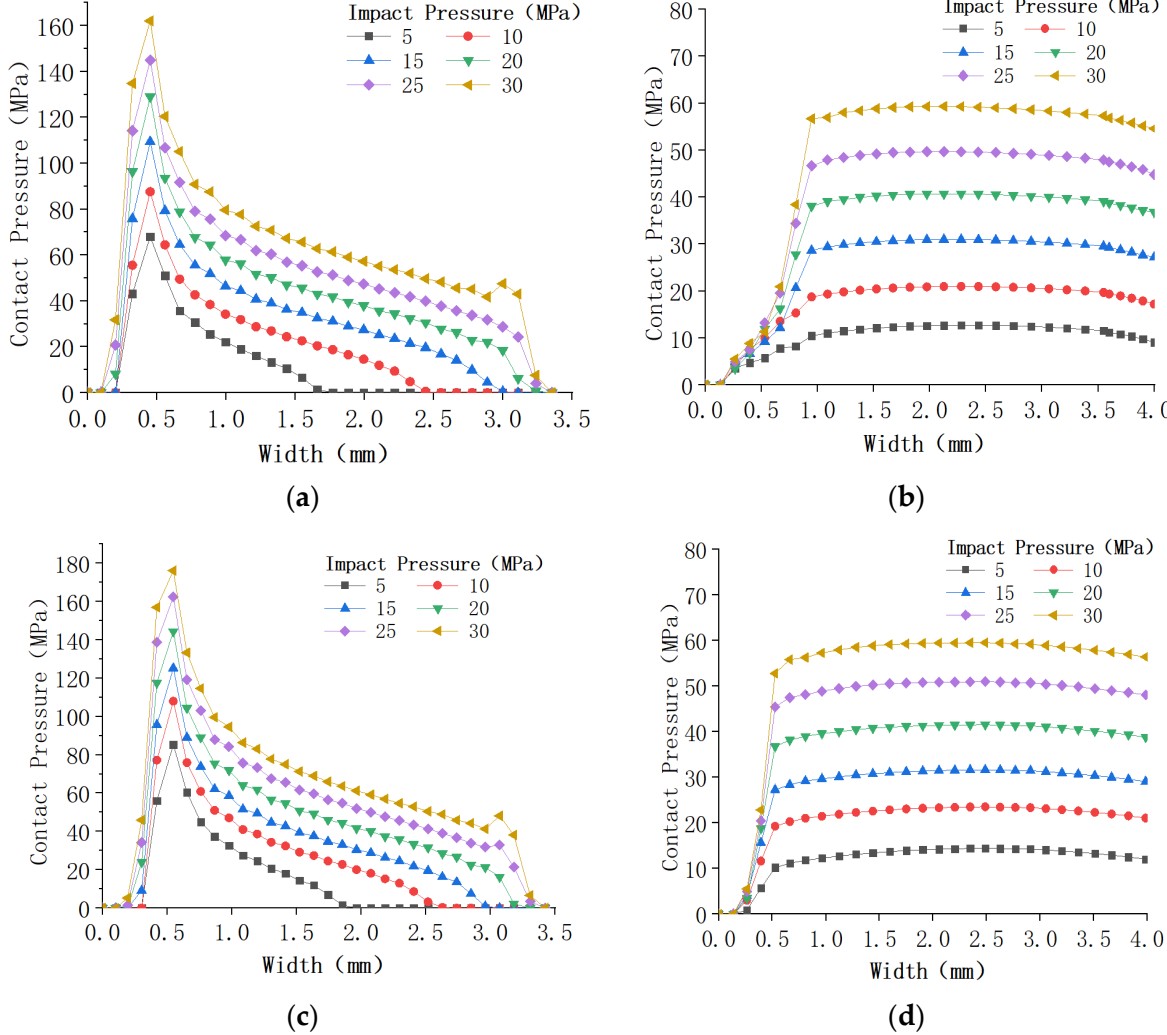

**Figure 14.** *Cont.*

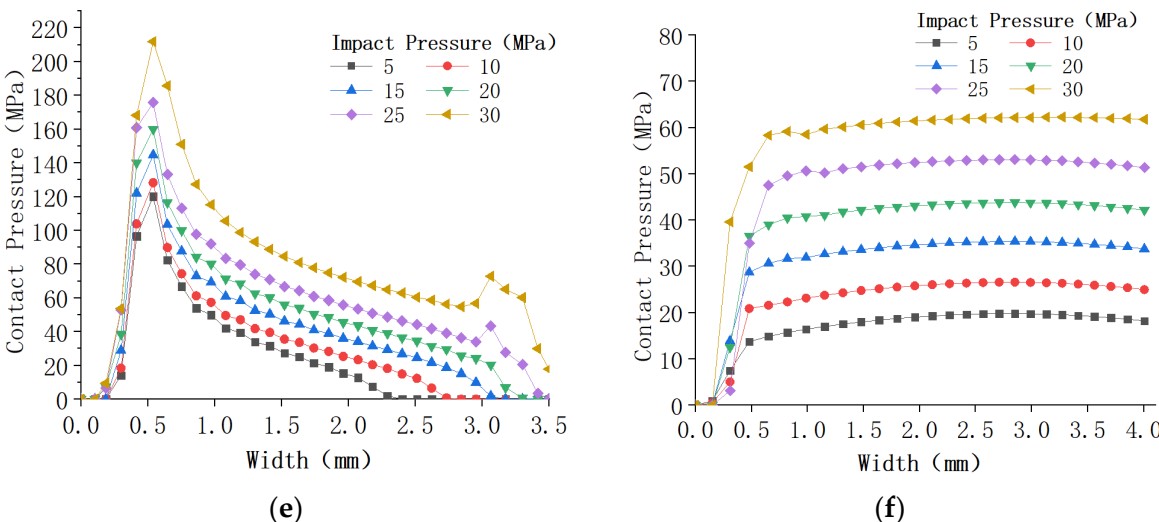

(**e**)　　　　　　　　　　　　　　　　　　　(**f**)

**Figure 14.** Contact pressure curves on sealing surfaces under different compression ratios and pressure impacts: (**a**) compression ratio 15%, sealing surface I, (**b**) compression ratio 15%, sealing surface III, (**c**) compression ratio 20%, sealing surface I, (**d**) compression ratio 20%, sealing surface III, (**e**) compression ratio 25%, sealing surface I, (**f**) compression ratio 25%, sealing surface III.

### 3.4. Experimental Results

After conducting testing experiments on two different compression ratio hydro-pneumatic springs using a hydro-pneumatic spring test bench, the following findings were made: during static loading tests, neither the 10% compression ratio hydro-pneumatic spring nor the 15% compression ratio hydro-pneumatic spring exhibited any oil leakage. However, during dynamic loading tests, the 10% compression ratio hydro-pneumatic spring experienced significant oil leakage. On the other hand, the 15% compression ratio hydro-pneumatic spring began to exhibit oil leakage after approximately 4600 reciprocating cycles in the test, leading to the termination of the experiment. Subsequent measurements were carried out to compare the wear and tear of the hydro-pneumatic spring sealing rings at different numbers of reciprocating cycles.

After subjecting GS sealing rings with a 15% compression ratio to a reciprocating loading test for varying numbers of cycles from left to right, specifically 1000, 2000, 3000, 4000, and 4600 cycles, we obtained five worn step seal rings, as shown in Figure 15. These step seal rings exhibited varying degrees of wear and deformation. The step seal ring after 4600 cycles of reciprocating testing suffered the most severe wear, with the lip completely worn flat, presenting an inclined surface overall. The maximum thickness of the cross-section for the five worn step seal rings was measured, as shown in Figure 16.

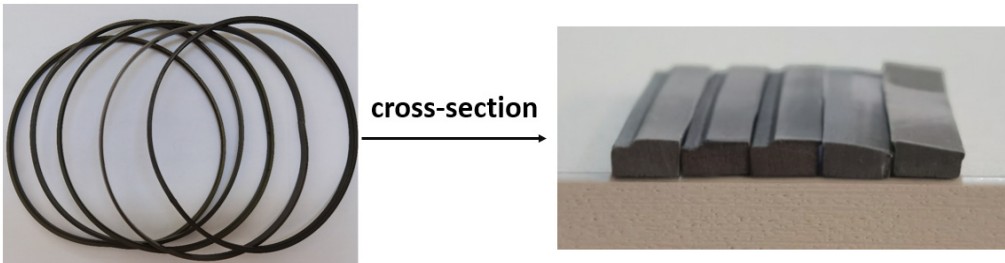

**Figure 15.** Wear condition of the step seal rings at different durations of use.

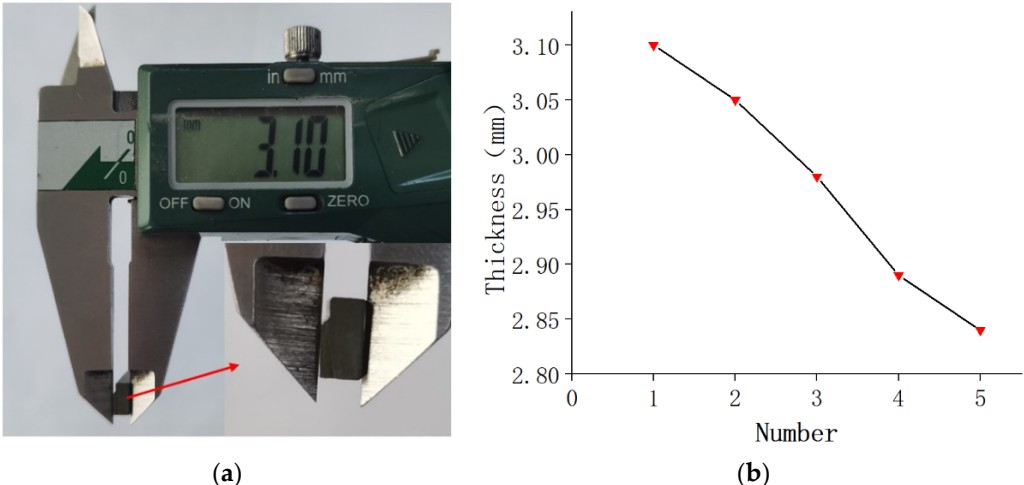

**Figure 16.** Maximum thickness of cross-section after step seal ring wear: (**a**) step sealing ring thickness measurement diagram, (**b**) thickness of stepped seal after wear.

## 4. Discussion

### 4.1. Compression Process of the Seal Ring

As shown in Figure 10, the compression ratio is directly proportional to the stress applied to the seal ring. When the compression rate is 10–20%, the stress generated by the sealing ring has a small influence range on the stepped ring. However, when the compression ratio reaches 25% to 30%, there is a noticeable increasing trend in the range of stress impact compared to a compression ratio of 20%. With the increase in compression ratio, the sealing width between the seal ring and the piston rod also increases. After compression, the maximum stress in the seal ring occurs at the contact surface between the step seal ring and the piston rod. However, when the compression ratio reaches 30%, the location of maximum stress shifts. It is no longer concentrated at the contact point between the step seal ring and the piston rod but shifts towards the inner portion of the step seal ring. This increases the risk of internal tearing of the step seal ring, and the impact of this failure mode is far more significant than the wear of the step seal ring.

The necessary condition for a seal ring to achieve sealing is that the maximum contact pressure on the sealing contact surface must be greater than or equal to the high hydraulic pressure. When comparing the contact pressure curves of the three sealing surfaces in Figure 11 and Table 2 at the same compression ratio, it is evident that, under the same conditions, the contact pressure on sealing surface I is significantly higher than that on sealing surfaces II and III. Therefore, the sealing pressure limit of the seal ring is determined by sealing surfaces II and III.

### 4.2. Pressure Impulse Process

From Figure 12, it can be observed that after the seal ring is subjected to pressure impulse, it moves towards the low-pressure side until it acts on the side of the seal groove at the bottom. With the increase in impulse pressure, the O-ring deforms in a concave shape. The reason for this is the friction occurring between the step ring and the O-ring. The interaction between the O-ring, the sealing groove, and the step ring hinders the movement of the O-ring, effectively reducing the risk of leakage due to changes in the relative position between the step seal ring and the O-ring. Under different compression ratios and pressure impulses, the deformation of the step seal ring is relatively small, while the O-ring undergoes significant changes in shape. This is determined by the material properties of both components. This reciprocating sealing arrangement utilizes this property of the O-ring to achieve sealing under pressure impulses.

From Figure 13, it can be seen that when the compression ratio is 10% and the seal ring is subjected to pressure impulse, there is a severe misalignment between the O-ring and the



step seal ring. Moreover, after the pressure impulse ends, they do not return to their original positions. In practical use, when the seal ring encounters significant pressure impulses, it can result in the displacement or even twisting of the seal ring, causing irreversible damage. This situation should be avoided. When the compression ratio reaches 15% or higher, the severe misalignment between the O-ring and the step seal ring disappears. This is because as the compression ratio increases, when it exceeds 15%, the strain on both the step seal ring and the O-ring increases, and the contact surface between them widens. This leads to a more even distribution of forces, better repetitive positioning, and effectively prevents leaks caused by the movement of the seal ring.

*4.3. Test Process Analysis*

The results of static loading tests indicated that when the inflation pressure of the hydro-pneumatic spring was around 7 MPa, there was no leakage observed in either of the two hydro-pneumatic springs with seal ring compression ratios of 10% and 15% under static loading conditions.

Through dynamic loading tests, it was found that the internal pressure of the hydro-pneumatic spring reached approximately 23 MPa at maximum working stroke. The hydro-pneumatic spring with a seal ring compression ratio of 10% experienced hydraulic fluid leakage during the initial reciprocating motion of the hydro-pneumatic spring and stopped the follow-up trial. The hydro-pneumatic spring with a seal ring compression ratio of 15%, on the other hand, did not exhibit significant leakage in the first 4000 cycles of reciprocating motion testing. However, at approximately 4600 cycles, the seal ring began to leak oil. As shown in Figure 16, at this point, the lip of the step seal on the seal ring had been completely worn flat, presenting a sloped condition, and the seal ring had lost its sealing capability entirely.

Building upon the analysis in Section 4.1, when the maximum stress is concentrated on the contact surface between the step seal ring and the piston rod, in the practical operation of the hydro-pneumatic spring, as the piston rod reciprocates, there is repeated friction between the piston rod and the step seal ring. The greater the stress on the step seal ring, the greater the frictional wear it receives. A Figure 15 depicts the wear conditions of the step seal ring under different durations of use. Due to the prolonged reciprocating friction between the step seal ring and the piston rod, the contact portion of the step seal ring with the piston rod experiences severe wear, eventually resulting in the flattening of the step seal ring lip, causing the seal ring to lose its sealing performance [22]. The maximum cross-sectional thickness of the step seal ring after testing with five sets of seal rings is shown in Figure 16. The step seal rings suffered varying degrees of wear and deformation. The two seal rings with the most severe wear had their step seal ring lips completely worn flat, presenting an overall sloped condition. Although there was no oil leakage at 4000 cycles, by this point, the lip of the step seal ring had been completely flattened. At this stage, the contact surface width was excessive, and even if it could still maintain a seal under this pressure impulse, the resulting friction would limit the damping characteristics of the hydro-pneumatic spring and impact its performance.

The tests, which included both static and dynamic loading tests, verified the sealing performance of the hydro-pneumatic spring seal rings under two different compression ratios. To a certain extent, these tests confirmed the reliability of the simulations. The seal rings with compression ratios of 10% and 15% exhibited good sealing performance under static loading conditions. However, when subjected to impact pressure, the seal with a compression ratio of 10% experienced a change in its relative position, leading to leaks during subsequent impacts and a loss of sealing performance. On the other hand, when the seal ring had a compression ratio of 15%, it maintained good sealing performance even under impact pressure, ensuring that the seal surface did not leak. In subsequent sealing structure design and sealing ring selection, the appropriate sealing ring compression rate should be selected, and the wear resistance of the stepped ring should be improved to achieve the purpose of improving the service life of the sealing ring.

### 5. Conclusions

In this paper, the finite element model of the GS15006 seal ring was established by using ABAQUS finite element simulation software, and the working state of the hydro-pneumatic spring under impact load was simulated by using a pressure penetration module. The influence of different compression rates of the GS seal ring under reciprocating impact pressure on the stress and contact pressure of the seal ring was analyzed.

(1) For the GS seal ring used with the YQHF-CC180/150*200B hydro-pneumatic spring, the contact pressure on the sealing surface increases with the increase in the compression of the seal ring. The contact width of the sealing surface also increases with the increase in the compression of the seal ring. The maximum contact stress on the seal ring occurs at the lip of the step seal ring, and the maximum sealing pressure of the seal ring is determined by the contact pressure on sealing surfaces I and III.

(2) When the compression ratio of the seal ring is 10%, the seal ring shifts relative to its original position after being subjected to pressure impact, which can lead to oil leakage during use, as confirmed in experiments.

(3) By analyzing the contact pressure curves on sealing surfaces I and III under different compression ratios and pressure impacts, it was found that the contact pressure on the sealing surface is jointly determined by the compression and the impact pressure.

(4) Static and dynamic loading tests were conducted in experiments to verify the sealing performance of the hydro-pneumatic spring seal rings under two compression ratios. This validation confirmed the reliability of the simulation, which provides a reference for subsequent structural design and selection of seal parts for different types of hydro-pneumatic springs.

It should be pointed out that in the test verification to ensure the tightness of the hydro-pneumatic spring, we are currently unable to detect the contact pressure and contact width of the sealing surface of the seal when the hydro-pneumatic spring is working, which will continue to be studied in the future.

**Author Contributions:** Conceptualization, G.L. and Z.D.; methodology, S.W.; software, P.L.; validation, P.L.; formal analysis, Z.D. and G.L.; investigation, P.L. and Z.D.; data curation, D.L.; writing—original draft preparation, S.W.; writing—review and editing, S.W. and G.L.; supervision, D.L.; project administration, D.L. All authors have read and agreed to the published version of the manuscript.

**Funding:** This research was supported by the National Nature Science Foundation of China (Grant-No.52105054) and key scientific and technological projects in Henan Province (Grant-No.222102220012); all support is gratefully acknowledged.

**Institutional Review Board Statement:** Not applicable.

**Informed Consent Statement:** Not applicable.

**Data Availability Statement:** The data used to support the findings of this study are available from the corresponding author upon request.

**Conflicts of Interest:** The authors declare no conflict of interest.

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
