# Peer review of "Simulation and Experimental Study on Sealing Characteristics of Hydro-Pneumatic Spring GS Seal Rings"

_applsci, doi:10.3390/app132111703_

Round 1

Reviewer 1 Report

This study focuses on the crucial role of GS sealing rings in Hydro-pneumatic springs. These springs often face high pressure and alternating loads, leading to potential seal damage and leakage. The research emphasizes that dynamic seal failure is a primary cause of Hydro-pneumatic spring breakdowns. By examining various sealing structure parameters, the study aims to offer theoretical guidance for designing effective seals. Through simulations and experiments, it is found that the compression ratio of the sealing ring significantly affects its sealing performance. A compression ratio of 15% is identified as optimal, ensuring effective sealing up to 25MPa.

The study addresses a critical but often overlooked aspect of Hydro-pneumatic spring design – the sealing performance. This provides valuable insights for engineers and designers. The research uses finite element simulations and experimental validation to establish a solid theoretical basis for the design of GS sealing in Hydro-pneumatic springs. The findings offer practical guidelines for achieving optimal sealing performance, potentially leading to longer equipment life and enhanced reliability.

The paper could be accepted for publication after applying the following comments:

1.       Limited Application Scope: The study primarily focuses on the GS15006 seal ring and a specific model of Hydro-pneumatic spring (YQHF-CC180/150*200B). Further elaboration is required to clarify how the research findings can be applied to different models or types of seals.

2.       Lack of Real-time Monitoring: The study acknowledges a limitation in the current inability to monitor contact pressure and contact width of the sealing surface during operation. This section requires a more detailed explanation.

3.       Potential Material Sensitivity: The study provides material parameters for the GS seal ring based on single-axis compression tests. The sensitivity of these parameters to variations in material properties is not discussed, which could be a consideration in real-world applications. More explanation is needed.

Author Response

Dear reviwer,

Thanks very much for taking your time to review this manuscript. I really appreciate all your comments and suggestions!

We answered your questions and hope they will meet the requirements.

Question 1. Limited Application Scope: The study primarily focuses on the GS15006 seal ring and a specific model of Hydro-pneumatic spring (YQHF-CC180/150*200B). Further elaboration is required to clarify how the research findings can be applied to different models or types of seals.

In this paper, YQHF-CC15006/180*150B Hydro-pneumatic Spring is selected as the research object, and GS200 seal ring is selected as the main seal to study its sealing performance. When we need to design different types of Hydro-pneumatic Spring or replace seals from different types, we can use the research method in this paper to provide reference for the design of seal structure and the selection of seals. A description is added at the end of the article.

Question 2.  Lack of Real-time Monitoring: The study acknowledges a limitation in the current inability to monitor contact pressure and contact width of the sealing surface during operation. This section requires a more detailed explanation.

Because the contact pressure and seal contact width data involved in this paper are obtained from simulation, and these two parameters will affect the sealing performance.At present, we do not have a way to use experiments to compare simulation results.

Question 3.  Potential Material Sensitivity: The study provides material parameters for the GS seal ring based on single-axis compression tests. The sensitivity of these parameters to variations in material properties is not discussed, which could be a consideration in real-world applications. More explanation is needed.

After the experimental parameters of the material were imported into the software, the sealing performance simulation of the subsequent seals was completed by using the Mooney-Rivlin hyperelastic model. As shown in equation (1), I1 and I2 are invariants of the stress tensor, I3 and D reflect the degree of deformation of the material, and C10, C01 and D are constants of the material. It is calculated by importing the material test data of ABAQUS software.

Or please see the attachment.

Thanks very much for taking your time to review this manuscript. I really appreciate all your comments and suggestions!

Best regards,

Mr.Liu

Reviewer 2 Report

Dear authors,

for comments see the attached file.

Check for typos and punctuation

Author Response

Dear reviewer,

Thanks very much for taking your time to review this manuscript. I really appreciate all your comments and suggestions!

We answered your questions and hope they will meet the requirements.

Question1

“Title…”, GS … perhaps it would be better to avoid using acronyms in the title.

“Abstract…”, GS = Gland Seals ?

Reply: GS Seals come from the supplier of seals, which only represent the code name of this kind of seals, we can not get its specific meaning, not the meaning of Gland Seals.

Question2

Introduction

Line 40 … “it is inert gas as the elastic medium” … perhaps … it uses inert gas ?

Line 46 … “Thy dynamic” … the dynamic.

Line 48 to Line 50 … perhaps it would be good to introduce an explanatory figure.

Line 51 … “work” … works or operates.

Line 86 and Line 87 … “the shock load duration and vibration frequency of Hydro-86 pneumatic spring are long when working” … long vibration frequency ?? what does it mean.

Reply: Line 40…, We’ve changed [original text] to [edited text] (page 1, line 39).

Line 46 …, We’ve changed [original text] to [edited text] (page 2, line 45).

Line 48 to Line 50 …, We looked for a lot of literature, because each manufacturer's sealing materials and processes are different, we did not find those specific comparative data.

Line 51 … , We’ve changed [original text] to [edited text] (page 2, line 50).

Line 86 and Line 87…, We’ve changed [original text] to [edited text] (page 2, line 84 to line 86).

Question3

Sec.2.1

Line ??? … The dynamic seal … the dynamic seal.

Line ??? “The GS sealing ring is made of elastic force generated by the pre-compression of the O-ring” … can you improve the sentence?

Figure 3 … (a) and (b) are missing … please specify the units of measurement.

Reply:â‘ .Because the oil-gas spring has two types of sealing, it is necessary to make a difference.

â‘¡. We’ve changed [original text] to [edited text] (page 3, line 114 and line 115).

â‘¢. Figure 3…, We’ve changed [original text] to [edited text] (page 3, line 125).

Question4

Sec. 2.2

Line 135 to Line 147 … Why not show, as an example, some stress-strain curves obtained experimentally?

Reply: These curves were not added before because there are too many images in the article. After receiving your comments, we searched for these data again, because the experiment was conducted in April, and now we have not found the previous experimental data.

Question5

Sec. 2.2

it is not clear which calculation "model" was used: large displacements? large strains = finite strains?

please specify the selected calculation methods.

Line 200 … represent pressure penetration 200 surfaces … what does it mean.

Line 201 … please indicate in Figure 7 the end cap and the piston rod.

Line 210 … The compression ratios of the entire seal ring … please specify the technical definition of the “compression ratio”.

TABLE 1 … Sealn groove depth ? typo ? … compression ratio … please specify how this parameter is defined … Seal groove depth … indicate this dimension on the drawing.

Figure 8 … the pressure scale is absent.

Line 223 and Line 224 : check punctuation.

Line 227 … Why is a capital letter used after the colon?.

Line 227 … The pressure penetration method … see comment on Line 200.

Line 237 … 5 MPa to 30 MPa … are these the peaks in the graph in Figure 8? improve the correspondence between what is reported in the text and what is indicated in the figure.

Line 255 … why “:” ?.

Line 257 … compression rate ? … or ratio ? please specify how this parameter is defined.

Line 259 … compression rate ? … or ratio ? please specify how this parameter is defined.

Line 270 … test bench, Let it … check the punctuation.

Figure 10 … Line 306 to Line 308 … the letters (a) (b) (c) (d) (e) are missing … units for stresses?

what happens to the strains? it would be good to have at least one image with the results of the equivalent strain.

TABLE 2 … pressure units ???

Figure 11 … (a) and (b) are missing.

Line 351 … why capitalized?

Line 351 to Line 357 … it is necessary to indicate in the Figure 12 which and where the stress values referred to in the text are.

Figure 12 … for each row … what is the parameter that is varying from left to right in figure 12?

Line 369 … stress impact ?

Reply: What we want to express here is that rubber will produce large deformation and large displacement after being stressed, and this result will lead to non-convergence of the software in the analysis. The calculation model used in the software simulation is the Mooney-Rivlin superelastic model. This model is explained in Sec.2.2.2.

Line 200 …, I did not see "represent pressure penetration ‘200’ surfaces" in the original text, perhaps because of the display problem.

Line 201…, We have made changes in Figure 7.

Line 210… and TABLE 1 … , Additions and explanations have been made. The compression rate of the sealing ring is controlled by the depth of the sealing groove. We added the sealing groove depth in Figure 2.

Figure 8 …, The pressure value P here is a variable and is described in 2.3.4.

Line 223 and Line 224, Line 227 … , Line 227 …, We have made changes.

Line 237 …, We’ve changed [original text] to [edited text] (page 6, line 252 and line 254).

Line 255 …, Line 257 …, Line 259 …, Line 270 …, Figure 10…, We have made changes. In this section, we only judge the location of the possible damage of the seal by the stress concentration phenomenon, So there is no picture of the equivalent strain.

TABLE 2 …, We’ve changed [original text] to [edited text] (page 6, line 353).

Figure 11…, Line 351…, We have made changes.

Line 351 to Line 357 …, In the first image of each row, we add a unified ruler of the cloud image of this row, and we can get the stress value by comparing the color of the ruler.This will make the picture more concise.

Figure 12 …, We’ve changed [original text] to [edited text] (page 9, line 369).

Line 369 …, Here refers to the seal stress cloud map after the pressure impact, the form of the pressure impact is shown in Figure 8.

Question6

Sec. 3.2.2

Results shown in Figure 13…are they static analysis results or time-dependent analysis results? It is not clear ? what history follows the "stress (pressure) impact"? the results refer to which time ? if time-dependent, the results refer to which value of the time variable? Are the simulated load conditions monotonic? Does the constitutive model (and your analysis) take into account the non-linear unloading phases of the stress-strain curve? ... I do not believe …

Reply: Figure 13 shows the stress cloud diagram of the seal ring after the 15MPa pressure impact, which has been added in the paper. We’ve changed [original text] to [edited text] (page 10, line 390 to line 393). Here, the analysis results depend on time. Since the pressure impact has only been carried out for two cycles, the change analysis after two cycles has only been carried out at present. The main focus is to study the shape of the seal after the pressure impact and observe whether the seal can return to the initial compression state.

Question7

Sec. 3.4

Line 425 … capital letters after the colon! … check this “practice” throughout the text.

Figure 15 … In my opinion this is not only “wear” but especially the effects of residual strains in the material of the seal ring. Did you weigh the rings at the end of the test? if there was any wear you should find rings that reduce their weight … add some comments to this consideration at least in the conclusions.

Reply: Line 425 …, We’ve changed [original text] to [edited text] (page 6, line 425).

Thank you very much for your suggestions, these comments are very helpful to us, we will use this as a measure of sealing ring test in the future.

Thanks very much for taking your time to review this manuscript. I really appreciate all your comments and suggestions!

Best regards,

Mr.Liu